# Targeted Neonatal Echocardiography in Bronchopulmonary Dysplasia: A Framework for Screening and Management of Chronic Pulmonary Hypertension

**DOI:** 10.3390/jcm14228161

**Published:** 2025-11-18

**Authors:** Audrey Hébert, Andréanne Villeneuve, Anie Lapointe, Christine Drolet, Nina Nouraeyan, Brahim Bensouda, Carolina Michel-Macias, Laila Wazneh, Marco Zeid, Floriane Brief, Gabriel Altit

**Affiliations:** 1Division of Neonatology, CHU de Québec, Université Laval, Quebec City, QC G1V 0A6, Canada; audrey.hebert.2@ulaval.ca (A.H.); christine.drolet.med@ssss.gouv.qc.ca (C.D.); 2Division of Neonatology, CHU Sainte-Justine, Université de Montreal, Montreal, QC H3T 1C5, Canada; andreanne.villeneuve.med@ssss.gouv.qc.ca (A.V.); anie.lapointe@umontreal.ca (A.L.); floriane.brief@gmail.com (F.B.); 3Division of Neonatology, Jewish General Hospital, McGill University, Montreal, QC H3T 1E2, Canada; nina.nouraeyan@mcgill.ca; 4Division of Neonatology, Hôpital Maisonneuve-Rosemont, Université de Montreal, Montreal, QC H1T 2M4, Canada; brahim.bensouda@umontreal.ca; 5Division of Neonatology, Montreal Children’s Hospital, McGill University Health Centre, McGill University, 1001 Decarie, Montreal, QC H4A 3J1, Canada; dra.carolinamichel@gmail.com (C.M.-M.); laila.wazneh@muhc.mcgill.ca (L.W.); marco.zeid@muhc.mcgill.ca (M.Z.)

**Keywords:** chronic pulmonary hypertension, bronchopulmonary dysplasia, prematurity, chronic lung disease

## Abstract

Chronic pulmonary hypertension (cPH) associated with bronchopulmonary dysplasia (BPD) is a major contributor to morbidity and mortality in extremely preterm infants. Despite improvements in neonatal care, the burden of BPD and its pulmonary vascular complications remains significant. Early detection and standardized management of cPH are essential to improve outcomes. Echocardiography plays a central role in screening and guiding treatment, particularly in high-risk infants requiring respiratory support at or beyond 36 weeks postmenstrual age. The Targeted Neonatal Echocardiography—Quebec (TnECHO-Qc) collaborative has developed a province-wide screening and management algorithm for cPH in preterm infants with BPD. This initiative outlines a stepwise approach to echocardiographic evaluation, including specific criteria for identifying elevated pulmonary arterial pressures, grading severity, and scheduling follow-up based on clinical and imaging findings. Additional management elements encompass biomarker use, respiratory and nutritional optimization, and consideration of airway anomalies, reflux, and aspiration. Pharmacologic therapies, including inhaled nitric oxide and pulmonary vasodilators, are considered for moderate to severe cPH with a pre-capillary component (“pulmonary arterial hypertension”) after stabilization of ventilation and oxygenation, and guided by echocardiography follow-up. This collaborative initiative establishes a standardized, multidisciplinary framework to enable timely recognition and individualized management of chronic pulmonary hypertension (cPH) in preterm infants. The primary goal is to reduce adverse outcomes and support long-term health, with the effectiveness of the framework to be evaluated through longitudinal outcome assessments.

## 1. Introduction

Chronic pulmonary hypertension (cPH) associated with bronchopulmonary dysplasia (BPD) presents a significant challenge in the care of preterm infants. Despite advances in neonatal care, the incidence of BPD remains high among extremely preterm infants, and its association with cPH contributes to substantial morbidity and mortality in this population [1]. Furthermore, diagnosis of cPH is challenging since clinical presentation is variable and could be confounded by underlying lung disease. Echocardiography has emerged as a valuable tool for early detection and monitoring of cPH in preterm infants with BPD, enabling timely detection in order to prevent worsening and improve outcomes [2].

In recent years, scientific societies such as the American Heart Association (AHA), the American Thoracic Society (ATS), and the Pediatric Pulmonary Hypertension Network (PPHNET) have published recommendations emphasizing a collaborative approach to early detection and appropriate treatment of patients with BPD affected by cPH [2,3]. The Targeted Neonatal Echocardiography—Quebec (TnECHO-Qc) Collaborative is a province-wide group of neonatologists with expertise in neonatal hemodynamics, representing neonatal units that care for preterm infants across Quebec, Canada. The collaborative aims to standardize clinical practice by developing unified care bundles and organizing joint academic meetings, fostering a coordinated, evidence-based approach to neonatal cardiovascular care at the provincial level. By integrating existing international recommendations with updated evidence and harmonizing practices across Quebec, this framework aims to provide clinicians with a practical tool to standardize care and ultimately improve outcomes for this high-risk population. As such, our group attempted to address the complexities of cPH in the context of BPD and delineate evidence-based strategies to mitigate its impact on neonatal health through a standardized approach. This manuscript describes this collaborative initiative that led to a province-wide standardized screening and management strategy for cPH associated with BPD in preterm infants, with a focus on the role of echocardiography in guiding clinical decision-making.

### Framework Development

The TnECHO-Qc collaborative is a multidisciplinary group focused on advancing neonatal hemodynamic care in the province of Quebec, Canada. It includes six neonatologists trained in targeted neonatal echocardiography (TnECHO), two neonatal nurse practitioners, and neonatal hemodynamic fellows actively engaged in training, working in four level-3 NICUs in Quebec. The group meets regularly (8–10 times/year) to review current practices and develop consensus-based approaches to neonatal hemodynamics. Discussions within the group identified significant variability in the diagnosis and management of cPH in infants with BPD across the province. This observation led to a structured initiative aimed at developing a standardized, evidence-informed framework for the screening and management of BPD-associated cPH.

The framework development process (from October 2019 to December 2023) included a comprehensive literature review and a series of focused discussions around four pre-defined themes:Identification of the at-risk population for screening;Definition and echocardiographic grading of cPH severity;Follow-up strategies and the role of additional biomarkers;Pharmacological and non-pharmacological management strategies.

These domains were reviewed to align clinical practice with current evidence and promote consistency across participating centers. The consensus process involved all members of the TnECHO-Qc collaborative. Each domain was informed by a structured literature review, followed by iterative discussions during regular meetings. An agreement was reached through group consensus, with draft versions revised and circulated until all members approved the framework. Once consensus was reached, each representative brought back the draft protocol within their local institution and discussed it in collaboration with experts in neonatology, nursing, respiratory therapy, pulmonology, and cardiology. Feedback from these local implementation efforts was then shared with the TnECHO-Qc Collaborative, and proposed adjustments were reviewed during a second cycle to finalize recommendations for adoption across all participating centers. This work is presented as a framework development manuscript based on a literature review and expert consensus.

## 2. Targeted Population for Screening

Proposed screening criteria for cPH with echocardiography (Figure 1) are: infants born at <28 weeks’ gestation (up to 27^6^ weeks) who remain on respiratory or oxygen support at 36 weeks postmenstrual age (PMA) undergo screening echocardiography for chronic pulmonary hypertension (cPH) [2,3]. Additionally, screening is considered for infants born between 28 and 30^6^ weeks’ gestation if they meet one of the following criteria considered additional risk factors for cPH: birth weight <3rd percentile based on Fenton growth charts [4] or continued need for ventilatory and/or oxygen support at 36 weeks PMA.

Echocardiography screening is performed by neonatologists trained in TnECHO, or by pediatric cardiology if TnECHO expertise is unavailable. A standardized echocardiographic protocol was implemented across sites to ensure consistent data acquisition and interpretation.

In high-risk infants, earlier screening at 32 weeks PMA is considered. Indications for early screening include: extreme prematurity (<25 weeks’ gestation), birth weight <3rd percentile by Fenton growth charts, persistent need for invasive mechanical ventilation at 32 weeks PMA, or clinical features suggestive of pulmonary hypertension. These may include unexplained respiratory deterioration or signs of right heart failure, such as hypoxic spells non-responsive to oxygen, significant intermittent hypoxic events, sudden cardiovascular instability, peripheral edema, ascites, hepatomegaly, failure to thrive, and/or unexplained tachycardia.

## 3. Echocardiographic Markers of cPH and Definition

cPH in the context of BPD is a complex, multifactorial condition. It may arise from elevated pulmonary vascular resistance due to pulmonary arterial remodeling or a reduced vascular surface area due to alveolar simplification. Contributing factors can also include lymphatic abnormalities, anomalies in venous drainage, or progressive structural lesions such as pulmonary venous ostial stenosis or atresia [2]. Additionally, elevated pulmonary capillary wedge pressure (PCWP) or left atrial pressure—often secondary to a stiff or noncompliant left ventricle—can further exacerbate pulmonary hypertension, especially when there is concomitant arterial hypertension [2]. The presence of pre-tricuspid (e.g., atrial septal defect) and post-tricuspid (e.g., ventricular septal defect or patent ductus arteriosus) shunts may also influence pulmonary artery pressure by increasing pulmonary blood flow and/or transmitting systemic pressures to the pulmonary circulation. This complexity is further compounded by the heterogeneous nature of lung injury in BPD, where different regions of the lung may be variably affected. These overlapping pathophysiological processes contribute to the multifaceted and dynamic presentation of cPH in BPD, underscoring the need for individualized assessment and management strategies.

TnECHO enables comprehensive bedside assessment of cardiopulmonary function. It allows for the estimation of pulmonary artery pressure, evaluation of right and left ventricular systolic and diastolic function, identification of intracardiac and extracardiac shunts along with their hemodynamic impact, and assessment of pulmonary venous drainage and ventricular compliance. The current definition of pulmonary hypertension, characterized by abnormally elevated pressure in the pulmonary artery, was revised in 2022. It is now defined as a mean pulmonary artery pressure (mPAP) ≥20 mmHg at rest in individuals older than 3 months of age, confirmed by right heart catheterization (Table 1) [5].

As such, one key indicator on TnECHO is an estimated mean pulmonary arterial pressure (mPAP) ≥20 mmHg, derived from a pulmonary insufficiency jet (PIJ) assuming a right ventricular end-diastolic pressure (RVEDP) of 5 mmHg [6]. PIJ occurs during diastole, and its peak velocity reflects the estimated mean pulmonary artery to right ventricular (PA–RV) pressure gradient. The end-diastolic velocity provides an estimate of the diastolic PA–RV gradient. If either of these values yields an estimated pressure of ≥20 mmHg, the TnECHO should be flagged for concern regarding potential cPH.

While a strict cutoff for systolic pulmonary artery pressure (sPAP) has not been formally established as a diagnostic trigger for PH, most studies use a threshold of ≥40 mmHg at sea level as indicative of an underlying mean pulmonary artery pressure (mPAP) ≥ 20 mmHg. In pediatric populations, sPAP ≥ 40 mmHg is commonly cited as suggestive of PH [5]. This threshold is clinically relevant, as sPAP can be estimated noninvasively using the velocity of the tricuspid regurgitant jet (TRJ), more commonly obtained than the PIJ. A right ventricular–right atrial (RV–RA) pressure gradient ≥35 mmHg, combined with an assumed right atrial (RA) pressure of 5 mmHg, suggests elevated pulmonary pressures. However, this assumed RA pressure may underestimate true values in infants with cPH. Indeed, infants with cPH and elevated RV afterload often develop RV remodeling, which can impair ventricular compliance and lead to elevated RA pressures [7]. Accurate estimation of the TRJ-derived pressure gradient requires a complete and well-defined spectral Doppler envelope and the application of the modified Bernoulli equation (ΔP = 4 × [velocity]^2^).

Alternatively, when the PIJ or TRJ cannot be reliably measured, surrogate estimations of pulmonary artery pressure can be obtained using flow gradients across a restrictive ventricular septal defect (VSD) or a patent ductus arteriosus (PDA) [2]. These alternative Doppler measurements can provide valuable insight into pulmonary hemodynamics. Using the modified Bernoulli equation to estimate pressure gradients across a PDA should be approached with caution, as the fundamental assumptions of the equation—namely, a sudden acceleration of flow through a discrete, narrowed orifice—are not fully met. Nonetheless, the directionality and velocity profile of flow through the PDA across the cardiac cycle, when present, can yield valuable insights into the relationship between pulmonary and systemic vascular resistance (PVR/SVR) and the relative pressure difference between the systemic and pulmonary circulations. Patterns such as continuous left-to-right flow, bidirectional flow, or predominant right-to-left shunting may help infer the severity of PH and guide clinical interpretation in the absence of reliable PIJ or TRJ signals. In the presence of an unrestrictive PDA or VSD, systolic PA pressure will by definition be near-systemic due to pressure transmission and equalization. However, directionality may also provide information regarding the PVR/SVR relationship.

Finally, when an estimated sPAP is obtained through TnECHO, comparing it to the systemic systolic blood pressure (sBP) at the time of the TnECHO can help contextualize the severity of PH. This relative comparison is particularly useful when direct measurements are limited, equivocal, or influenced by technical or physiological factors. An sPAP approaching or exceeding systemic levels is highly suggestive of significant PVR and warrants close clinical attention.

In the absence of right ventricular outflow tract obstruction, the RV systolic pressure approximates sPAP. Similarly, when left ventricular outflow tract obstruction is absent, the LV systolic pressure mirrors aortic systolic pressure. Therefore, systolic flattening of the interventricular septum suggests that RV systolic pressure is approaching LV systolic levels [8]. More pronounced findings—such as septal flattening at peak systole (indicative of isosystemic PH) or septal bowing into the LV cavity (suggestive of suprasystemic PH)—are consistent with significantly elevated RV systolic pressures relative to LV systolic pressures. The systolic eccentricity index (sEI) provides a quantitative assessment of interventricular septal motion, with an sEI ≥1.3 serving as a marker of elevated RV systolic pressure and an indicator of PH [9]. Infants with a large unrestrictive post-tricuspid shunt that will pressure equalize the pulmonary and systemic compartments will, by definition, yield to septal flattening. More in-depth hemodynamic assessments may include evaluation of shunt directionality and estimation of the pulmonary-to-systemic flow ratio (Qp/Qs), which can help determine whether elevated PVR is contributing to the observed physiology. Bidirectional or right-to-left shunting, particularly in the presence of appropriate systemic blood pressure for age, may indicate a significant PVR burden.

Some of these infants may also exhibit concomitant systemic hypertension, often due to elevated SVR in the setting of prior acute kidney injury, increased vascular stiffness, BPD, or exposure to corticosteroids. Therefore, when assessing the RV/pulmonary arterial compartment in relation to the LV/aortic compartment, whether at the septal level or directionality of post-tricuspid shunts, it is essential to interpret findings within the broader context of systemic hemodynamics [10]. Accurate evaluation of PH severity must account for systemic blood pressure, as elevated systemic pressures may mask or alter the perceived pressure gradients across shunts or valves, potentially leading to underestimation or mischaracterization of pulmonary vascular disease. Similarly, a patient may have elevated pulmonary arterial and RV systolic pressures, but have a round LV at the peak of systole due to systemic hypertension [10]. Doppler assessments of the RVOT and LVOT will provide valuable information, such as the possibility to estimate stroke distance using the velocity time integral, and assess the pulmonary acceleration time to RV ejection time.

In addition, echocardiographic evaluation must include a comprehensive assessment of ventricular performance. Both RV and LV systolic and diastolic properties should be systematically evaluated using a combination of conventional and advanced parameters. LV function can be assessed using measures such as fractional shortening (FS) and ejection fraction (EF), while right ventricular systolic function is commonly evaluated using tricuspid annular plane systolic excursion (TAPSE) and right ventricular fractional area change (FAC). For both ventricles, tissue Doppler imaging (TDI) provides key information on longitudinal myocardial velocities—S′ (systolic), E′ (early diastolic), and A′ (late diastolic). Additionally, myocardial strain analysis offers an assessment of global and regional ventricular performance. These parameters provide an integrated view of biventricular function, which is essential in the context of PH and evolving cardiopulmonary disease.

Additional echocardiographic considerations include confirming normal pulmonary venous return by color and PW-Doppler pattern, detection of outflow tract obstruction, and screening for anomalies of the great vessels (such as stenosed or dilated pulmonary arteries). TnECHO will also be used to measure cardiac dimensions in order to evaluate if there are signs of ventricular dilatation or hypertrophy.

When evaluating preterm infants, each echocardiogram performed as part of the suggested recommendations should include a detailed assessment of the pulmonary vein ostia using color Doppler with a low-velocity filter, alongside pulsed-wave Doppler interrogation of each vein to characterize normal biphasic or triphasic low-velocity flow patterns. It is important to note that in cases of pulmonary venous atresia, echocardiography may fail to detect the abnormality due to the complete absence of flow in the affected vein. In contrast, significant (especially multi-vessel) pulmonary vein stenosis is often associated with signs of pulmonary edema on chest radiography, reduced left atrial preload, and hemodynamic consequences such as exacerbation of right-to-left interatrial shunting or reduced LV preload and output (particularly in the absence of an interatrial communication) [11].

Beyond the estimation of pulmonary artery pressures, assessment of ventricular function, and evaluation of shunt physiology, TnECHO enables detailed phenotyping of the pulmonary circulation. This includes distinguishing whether the predominant pathology lies within the pulmonary arterial compartment, involves adverse pulmonary venous drainage, or reflects the impact of LV diastolic dysfunction, such as increased stiffness, on pulmonary venous return. As outlined in the diagnostic algorithm developed by our group, certain clinical scenarios—particularly those with severe disease, diagnostic uncertainty, or lack of clinical improvement—require multidisciplinary discussion involving neonatology, cardiology, and respiratory medicine. In these complex cases, invasive hemodynamic assessment via cardiac catheterization may be necessary to clarify the underlying physiology and guide appropriate management decisions.

## 4. Grading PH Severity via Echocardiography

Our group adhered to current recommendations [3] for grading the severity of cPH and categorized cases as mild, moderate, or severe based on a combination of TnECHO findings and clinical indicators (Table 1). While prior guidelines offered limited guidance on the specific metrics used for severity stratification, we developed a standardized set of criteria to support consistent and objective risk assessment. Although many infants with cPH may present with subtle or nonspecific clinical signs, severe cases are often associated with overt manifestations, such as hypoxemia (when an intracardiac, extracardiac, or intrapulmonary right-to-left shunt is present), hepatomegaly, hypotension, and growth failure. This reflects significant hemodynamic compromise and an increased risk of life-threatening complications [12]. These clinical features also highlight the critical importance of early recognition, comprehensive evaluation, and timely intervention to mitigate adverse outcomes and optimize long-term prognosis.

## 5. Echocardiographic Follow-Up

In the context of cPH screening in preterm infants, a structured and timely approach is essential for effective detection and management. All infants who meet screening criteria should undergo a comprehensive echocardiographic assessment by 36 weeks PMA. Our group’s recommendations include the consideration for early screening at 32 weeks PMA when risk factors are present, with repeat evaluation at 36 weeks PMA. Repeat evaluations are also advised for infants who continue to require supplemental oxygen or respiratory support at 40 weeks PMA, or earlier if there is evidence of clinical deterioration. For infants who were not screened at 36 weeks but later present with new-onset oxygen dependency or respiratory support, a prompt echocardiographic evaluation is recommended. In any of these scenarios, if the screening TnECHO is suggestive of moderate to severe cPH, a comprehensive multidisciplinary evaluation should follow. These include consultation with cardiology and respiratory medicine, as well as consideration for other professionals, such as otorhinolaryngology, occupational therapy/speech language pathology (for feeding and aspiration assessments), and, sometimes, genetics. This assessment is strongly recommended prior to discharge to ensure coordinated care and appropriate long-term follow-up.

Follow-up echocardiography should be planned based on cPH severity, at time intervals between 2 and 4 weeks [13,14,15]. For infants without PH on the screening TnECHO at 36 weeks PMA, a follow-up is considered at 40 weeks PMA only if the patient remains on oxygen and/or ventilatory support. For patients with a “mild” cPH, a follow-up is considered one month later, and may be scheduled as an outpatient evaluation in cardiology to avoid delaying discharge. Moderate cPH cases necessitate closer follow-up, with echocardiography performed every 1 to 2 weeks during management initiation, transitioning to monthly assessments once stabilization or improvement is observed. Managing severe cPH requires close monitoring, with follow-up echocardiography guided by both clinical status, multidisciplinary discussions, and response to treatment, allowing timely adjustments to the management plan (Figure 1) [16].

## 6. Biomarkers for cPH Screening

When cPH is identified, obtaining a baseline measurement of pro-brain natriuretic peptide (proBNP) or N-terminal proBNP (NT-proBNP), where available, can be a useful adjunct to echocardiographic evaluation and a valuable tool for longitudinal monitoring [13]. These cardiac biomarkers reflect myocardial strain and may provide important insights into the hemodynamic status of infants with cPH, particularly in settings where access to TnECHO is limited. Elevated proBNP or NT-proBNP levels may suggest worsening cardiopulmonary status, and serial trends can help guide clinical decision-making. However, it is important to recognize that proBNP and NT-proBNP levels may fluctuate in response to non-pulmonary factors, including renal dysfunction, systemic hypertension, or other cardiovascular stressors. As such, absolute values alone may not be sufficient for diagnosis or risk stratification. Instead, significant upward trends or sudden increases in biomarker levels should prompt earlier echocardiographic reassessment to evaluate for evolving cardiac or pulmonary disease.

## 7. cPH Management

### 7.1. Non-Pharmacological Management

#### 7.1.1. Respiratory Management

In the presence of cPH, hypoxemia should be minimized by adjusting the saturation targets. Our group opted for targeting arterial oxygen saturation (SaO_2_) levels of 92–95% to optimize oxygenation [17]. Intermittent hypoxia can induce pulmonary vasoconstriction as an acute, adaptive response aimed at optimizing ventilation–perfusion matching by diverting blood flow away from poorly ventilated lung regions. However, in preterm infants or those with BPD, repeated hypoxic episodes can become maladaptive. This leads to endothelial dysfunction, pulmonary vascular remodeling, medial hypertrophy, and intimal fibrosis. Over time, these changes contribute to elevated PVR and the development or worsening of cPH [18].

Some infants with cPH may require respiratory support to achieve target pH levels of ≥7.30 and carbon dioxide (CO_2_) levels ≤60 mmHg [19]. Higher levels of positive end-expiratory pressure (PEEP) may be considered to optimize lung recruitment. For patients with BPD-cPH who still require mechanical ventilation, a ventilation strategy involving higher tidal volumes (e.g., 8–12 mL/kg) and lower frequencies (e.g., 20/min) may be beneficial [20]. However, overdistension by increasing mean airway pressure should be avoided, as it can compress the pulmonary capillaries within the alveolar walls, thereby contributing to elevated RV afterload. As part of discharge planning for patients with cPH, a baseline sleep oximetry evaluation should be considered to ensure ongoing respiratory stability and optimize long-term management, especially if home oxygen therapy is considered.

#### 7.1.2. Airway Lesions Screening

Patients with cPH are at increased risk of having both anatomical and dynamic airway lesions, including vocal cord movement anomalies, tracheomalacia, bronchomalacia, airway stenosis, and granulomas [19]. Infants with BPD-cPH should undergo an evaluation by otorhinolaryngology and pulmonology specialists if there is concern of airway involvement contributing to the patient’s clinical presentation. In cases where a rigid bronchoscopy is planned for further evaluation, cardiac anesthesia should be involved to ensure optimal management and safety during the procedure. This collaborative approach facilitates comprehensive assessment and management of airway complications in patients with cPH, optimizing global management and reducing potential risks associated with procedural interventions.

#### 7.1.3. Reflux and Aspiration Management

Gastroesophageal reflux (GER) and aspiration represent potential contributors to ongoing pulmonary and airway injuries in patients with cPH and BPD [21,22]. Therefore, it is important to consider assessing for GER and aspiration, as their presence can exacerbate respiratory symptoms and contribute to airway disease. If GER and aspiration are identified, multi-modal treatment should be considered as needed, following local guidelines and utilizing available resources, such as considering a consultation with gastroenterology in more complex cases [21]. This approach may mitigate further lung injury and optimize respiratory outcomes in patients with cPH, emphasizing the importance of a comprehensive management strategy addressing both pulmonary and gastrointestinal factors.

#### 7.1.4. Nutrition

Neonates diagnosed with cPH and BPD require a higher range of caloric intake, typically ranging from 130 to 150 kilocalories per kilogram per day (kcal/kg/d) [23]. Due to their increased metabolic demands, they are at risk of nutritional deficiencies, and close monitoring of nutrition and growth is essential. This includes regular assessments of harmonious weight gain and linear growth, as well as tracking body mass index and weight/length ratio, to ensure adequate nutritional support and promote optimal development, while avoiding excessive weight gain without corresponding linear growth, which may contribute to adverse cardiopulmonary outcomes [23]. Infants on mechanical ventilation, including those with tracheostomies and anticipated to transition to home ventilatory support, may have reduced caloric expenditure due to partial or full respiratory support. Additionally, many of these patients may have been exposed to repeated courses of corticosteroids for pulmonary inflammation. As a result, they are at risk of excessive weight gain, which can contribute to a restrictive chest wall pattern and adversely impact pulmonary and pulmonary vascular mechanics. These infants require close monitoring to ensure that growth remains proportional and appropriate, with nutritional support tailored to avoid overnutrition while promoting optimal development. Regular follow-up is essential to adjust nutritional strategies as needed and to support long-term respiratory and cardiovascular health, ideally in collaboration with a neonatal dietitian when resources are available. Of note, there are currently no universal standard guidelines for nutritional management in this patient population, highlighting the need for individualized care and multidisciplinary collaboration to address their complex medical needs (Table 2).

### 7.2. Pharmacological Management

At 36 weeks PMA, in neonates with moderate to severe cPH, a two-week trial of diuretics may be considered as an initial pharmacological intervention once ventilation and oxygenation have been optimized [24,25]. Baczynski M et al. have described an improvement in echocardiographic parameters of cPH after 3 days of furosemide, followed by two weeks of hydrochlorothiazide and spironolactone [24]. While diuretics can be used as adjunctive therapy in infants with BPD-associated cPH, recent evidence, including a 2024 meta-analysis, has not demonstrated consistent benefit, and their role should be considered with caution and individualized to the clinical context [26]. The ultimate choice of diuretics should be based on local protocols and guidelines. Electrolytes should be monitored if diuretic therapy is pursued for more than one week, as prolonged use in premature infants may be associated with potential adverse effects, including nephrocalcinosis, electrolyte imbalances, and impaired growth [26]. Clinical and echocardiographic response should be monitored, and further studies are needed to determine whether routine use of diuretics is beneficial in this population.

In cases of severe cPH with RV dysfunction (i.e., PH crisis), inhaled nitric oxide (iNO) should be considered and administered according to the local iNO protocol, unless there is a suspected significant component of post-capillary involvement. Response to iNO should be closely monitored clinically and with echocardiography. Furthermore, pulmonary vasodilators should be initiated in consultation with a pediatric cardiologist or PH specialist, particularly in the context of RV dysfunction and/or iso to supra-systemic PH, after optimizing all other aspects of pulmonary and airway care as described above. Sildenafil is suggested as a first-line pulmonary vasodilator, especially in infants who have shown a positive response to iNO [15]. However, sildenafil is not approved by the Food and Drug Administration but has been used in this population of patients to rescue RV function [27].

### 7.3. Systemic Hypertension

Some preterm infants, particularly those with BPD, may develop significant systemic hypertension (>95th percentile for PMA as per REF), which can contribute to LV remodeling [28]. This may result in increased LV stiffness and elevated LVEDP, which in turn can raise LA pressure and impair pulmonary venous drainage, exacerbating pulmonary vascular disease [29]. In these patients, early involvement of nephrology is recommended. An abdominal ultrasound with renal Doppler studies should be performed to assess for underlying renovascular abnormalities. Renal biomarkers and electrolyte status should be closely monitored using plasma and urine creatinine, urine electrolytes, and plasma electrolytes. Antihypertensive management should be tailored to the individual, with consideration for agents such as angiotensin-converting enzyme (ACE) inhibitors (e.g., enalapril) [10,30], as well as diuretics like hydrochlorothiazide or spironolactone.

## 8. Indications for Cardiac Catheterization

In the management of neonates with cPH, the consideration of cardiac catheterization for assessment and treatment is warranted in several specific scenarios [3]. These include suspicion of pulmonary vein stenosis, evaluation of shunting contribution to PH pathophysiology in the presence of a PDA or large inter-atrial or ventricular shunt, and when the condition is worsening or not improving despite ongoing management efforts. Additionally, cardiac catheterization may be indicated if supra-systemic pulmonary hypertension is detected at any point, but should be performed following rescue therapy (i.e., improvement of RV function and stabilization of cardiovascular status to tolerate the procedure). Cardiac catheterization should also be considered in the presence of LV functional abnormalities on echocardiography or abnormal dimensions (such as LV hypertrophy), or clinical concern of high LVEDP, such as chronic pulmonary edema on radiography, pulmonary effusion, or a rapid development of pulmonary edema upon attempt to improve PVR [3]. Similarly, evaluation and treatment may be warranted in cases where aortopulmonary collaterals are suspected. When cardiac catheterization is indicated, involving cardiac anesthesia is advised to optimize procedural safety and patient care.

## 9. Limitations

This framework also carries limitations inherent to consensus-based approaches. Certain recommendations are informed more by expert opinion than by high-quality evidence, and validation data are not yet available. The applicability of the protocol may be context-dependent and not directly generalizable outside of the Quebec neonatal network. In addition, specific components, such as the role of diuretics or the frequency of follow-up, are based on evolving evidence and may require revision as new data become available. Ultimately, feasibility, adherence, and clinical impact will need to be established through prospective implementation and evaluation.

## 10. Conclusions

cPH remains a significant complication of BPD in preterm infants, contributing to substantial morbidity and mortality. This review outlines a comprehensive, multidisciplinary framework developed by the TnECHO-Qc collaborative to guide consistent screening, diagnosis, and management of BPD-associated cPH across the province of Quebec neonatal units. Early identification and standardized management are essential to improve outcomes in this high-risk population. Key innovations include standardized echocardiographic definitions of cPH severity, consensus-based screening and follow-up strategies, and a multidisciplinary approach that combines respiratory, nutritional, and pharmacologic interventions. The framework is currently being implemented across participating centers, with formal evaluation of feasibility, adherence, and impact underway. Future work will focus on validating this approach and generating outcome data to inform best practices and guide further refinements.

## Figures and Tables

**Figure 1 jcm-14-08161-f001:**
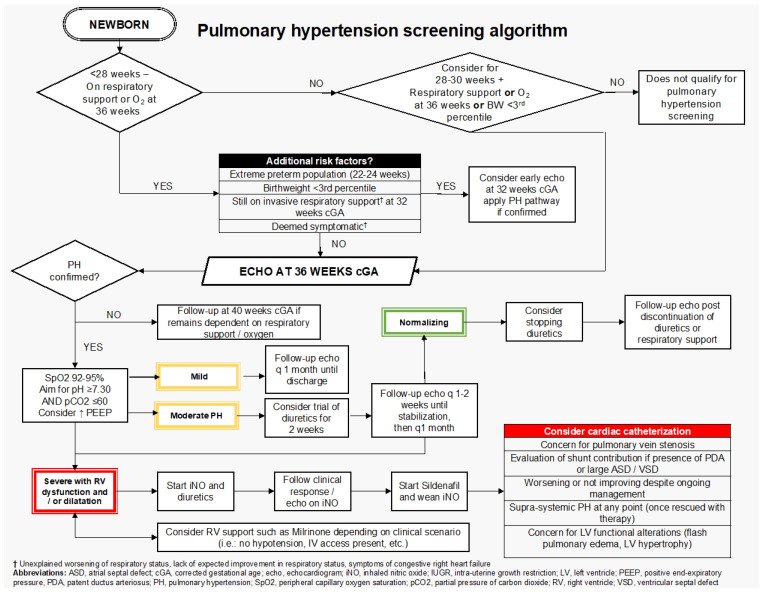
cPH screening algorithm.

**Table 1 jcm-14-08161-t001:** Pulmonary hypertension definition and severity grading.

PH Severity	
No PH	Right Ventricular systolic pressure less than 1/3 systemic pressure by tricuspid regurgitant jet (TRJ) or other metric (VSD, PDA); septal position round; LV eccentricity index less than 1.3; no RV hypertrophy; normal RV size and function
*Definition of pulmonary hypertension* mPAP > 20 mmHg (pulmonary insufficiency jet) sPAP > 40 mmHg (TRJ > 35 mmHg; VSD/PDA gradient) *Concerns of pulmonary hypertension* PAAT/RVET less than 0.25 (RVET/PAAT greater than 4) Eccentricity index greater than 1.3 Septal flattening at peak systole D-RV/D-LV greater than 1.00 Inter-atrial or post-tricuspid shunt with a right-to-left directionality Signs of pulmonary venous stenosis	
Mild	RV systolic pressure 1/3–1/2 systemic pressure; septal flattening in systole, RV function normal
Moderate	RVSP ½–2/3 systemic pressure; septum flattening in systole, RVH or dilatation, RV with altered function (TAPSE 6.5 to 8 mm; FAC: 20–30%): - TAPSE (Abnormal with <Zscore −2) - FAC < 30%
Severe	RV systolic pressure greater than 2/3 systemic pressure; If present, shunt (inter-atrial, post-tricuspid) with predominant R-L gradient, septal bowing, RVH, severe RV dysfunction, RV dilatation. Dilated right atrium and dilated inferior vena cava are also evidence of RA hypertension and RV diastolic dysfunction. Some concerning signs for severe alteration of RV function, TAPSE < Z score −3 Pericardial effusion FAC less than 20% Clinical signs of concern for severe RV dysfunction: Hypoxia secondary to R->L shunting to unload RA Hepatomegaly Hypotension Growth failure

**Table 2 jcm-14-08161-t002:** Overview of cPH management.

Domain	Key Recommendations
Respiratory Management	Target SaO_2_ 92–95%; Maintain pH ≥ 7.30 and PaCO_2_ ≤ 60 mmHg; Consider higher PEEP; Use higher tidal volumes (8–12 mL/kg) and lower rates; Baseline sleep oximetry before discharge.
Airway Lesion Screening	Evaluate for vocal cord dysfunction, tracheomalacia, bronchomalacia, and stenosis; Refer to ENT if airway involvement is suspected; Involve cardiac anesthesia for bronchoscopy.
Reflux and Aspiration Management	Assess for GER and aspiration; Initiate treatment as needed based on local protocols to reduce further lung injury and optimize respiratory status.
Nutrition	Provide 130–150 kcal/kg/day; Monitor growth closely Involve a neonatal dietitian Individualize care due to a lack of standardized nutritional guidelines
Pharmacological Management	Trial of diuretics at 36 weeks PMA in moderate to severe cPH after stabilization; Monitor electrolytes if >1 week; Consider iNO and sildenafil in severe cases with RV dysfunction; Use vasodilators under specialist guidance (neonatal hemodynamics and pediatric cardiology).
Cardiac Catheterization	Indicated for suspected pulmonary vein stenosis, significant shunting, or lack of improvement; Consider if supra-systemic PH or LV dysfunction present; Cardiac anesthesia is recommended for procedural support.

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
