# Peer review of "Targeted Neonatal Echocardiography in Bronchopulmonary Dysplasia: A Framework for Screening and Management of Chronic Pulmonary Hypertension"

_jcm, 2025, doi:10.3390/jcm14228161_

Round 1
Reviewer 1 Report
Comments and Suggestions for Authors
This manuscript presents a comprehensive framework for the screening and management of chronic pulmonary hypertension (cPH) in preterm infants with bronchopulmonary dysplasia (BPD), developed by the TnECHO-Qc collaborative. The topic is clinically relevant and addresses an important gap in neonatal care. However, the article lacks original data support, the methodology is vague, and the structure is loosely organized, all of which affect the scientific rigor and clarity.
1. Lack of Novelty and Original Data
The proposed framework is largely a compilation of existing recommendations from AHA, ATS, and PPHNet. The manuscript does not make clear what new knowledge or innovative approach it contributes. The most unique aspect is its origin from a provincial collaborative (TnECHO-Qc), yet no data is provided to show that this framework has been implemented, tested, or found to be feasible or effective in any way.
- Unclear and Underdescribed Methodology
The process by which this consensus was reached is not transparent. The reader is not informed about the number of experts involved, the literature review strategy, the methods for discussion and voting, or the criteria for reaching agreement. This is a critical flaw for any work purporting to be a consensus statement.
- Structural and Organizational Issues
The manuscript is overly long and repetitive, particularly in the sections on pathophysiology and echocardiographic markers. The flow between sections is often disjointed.
- The recommendation of the American Thoracic Society (ATS) and the American Heart Association (AHA) regarding the use of supplemental oxygen therapy for bronchopulmonary dysplasia associated pulmonary arterial hypertension is to control the target blood oxygen saturation (SaO2) between 92% and 95%. The recommended recommendation in this article is to maintain the SaO2 level of cPH patients at ≥92-95%, which is not a specific range. Clinically, there may be ambiguity regarding the low line value of SaO2 (i.e. choosing 92% or 95%).
- The conclusion is vague. It should precisely summarize the framework's key innovations and explicitly state the next steps for validation.
While the intent of your work is valuable, the manuscript in its present form does not meet the bar for original research or a definitive consensus statement. I strongly encourage you to gather preliminary data on the application of your framework within the TnECHO-Qc network. Submitting a manuscript that demonstrates the real-world feasibility, acceptability, and initial outcomes of your algorithm would be a much stronger contribution to the literature.
Author Response
October 30th, 2025
Dear Reviewers,
We would like to thank you for the careful evaluation of our manuscript entitled “Targeted Neonatal Echocardiography in Bronchopulmonary Dysplasia: A Framework for Screening and Management of Chronic Pulmonary Hypertension.”
We are grateful for the insightful comments and constructive suggestions provided, which have allowed us to clarify, refine, and strengthen our work. In this revised version, we have addressed all reviewer comments point by point and made changes to highlight the originality, methodological transparency, and clinical relevance of the proposed framework.
We appreciate the opportunity to improve our manuscript and look forward to your consideration.
Sincerely,
Dr Gabriel Altit, on behalf of the co-authors.
Dr Gabriel Altit, MDCM, MSc, FRCPC, FAAP
Neonatology – Montreal Children’s Hospital - McGill University
gabriel.altit@mcgill.ca
Reviewer 1
This manuscript presents a comprehensive framework for the screening and management of chronic pulmonary hypertension (cPH) in preterm infants with bronchopulmonary dysplasia (BPD), developed by the TnECHO-Qc collaborative. The topic is clinically relevant and addresses an important gap in neonatal care. However, the article lacks original data support, the methodology is vague, and the structure is loosely organized, all of which affect the scientific rigor and clarity.
- Lack of Novelty and Original Data
The proposed framework is largely a compilation of existing recommendations from AHA, ATS, and PPHNet. The manuscript does not make clear what new knowledge or innovative approach it contributes. The most unique aspect is its origin from a provincial collaborative (TnECHO-Qc), yet no data is provided to show that this framework has been implemented, tested, or found to be feasible or effective in any way.
Response: We appreciate this comment. Our framework is not intended to replace AHA, ATS, or PPHNet guidelines, but rather to merge their recommendations into a single, pragmatic tool for NICU application. We also integrated new evidence published since these guidelines and provide consensus from a provincial collaborative (TnECHO-Qc) to ensure feasibility across multiple centers. Publishing this provincial framework may be useful for other units seeking to harmonize practice. Importantly, existing guidelines do not provide detailed guidance on the process of implementation, nor do they offer practical instructions for classification or the use of echocardiographic metrics in clinical workflows. Our framework addresses these gaps by outlining actionable steps for bedside application. While implementation data are still in progress, we hope to publish these results as follow-up work building on this framework.
We have audits of the protocol’s implementation going on in the units. However, this data could not be incorporated into the manuscript, as it originates from a Quality Improvement (QI) initiative for which Research Ethics Board (REB) approval was not obtained for publication of the numerical findings within a manuscript.
Unclear and Underdescribed Methodology
The process by which this consensus was reached is not transparent. The reader is not informed about the number of experts involved, the literature review strategy, the methods for discussion and voting, or the criteria for reaching agreement. This is a critical flaw for any work purporting to be a consensus statement.
Response: We thank the reviewer for raising this point. We have revised the manuscript to clarify the methodology. The consensus process involved all members of the TnECHO-Qc collaborative, with each domain informed by a structured literature review and discussed iteratively during regular meetings. Draft versions were circulated for feedback and refined until unanimous agreement was reached. This structured and iterative approach is consistent with accepted methods for consensus development in small expert groups. Once consensus was reached, each representative brought back the draft protocol within their local institution and discussed in collaboration with experts in neonatology, nursing, respiratory therapy, pulmonology, and cardiology. Feedback from these local implementation efforts was then shared with the TnECHO-Qc Collaborative, and proposed adjustments were reviewed during a second cycle to finalize recommendations for adoption across all participating centers. These details were added to the methodology description.
- Structural and Organizational Issues
The manuscript is overly long and repetitive, particularly in the sections on pathophysiology and echocardiographic markers. The flow between sections is often disjointed.
Response: We thank the reviewer for this valuable feedback. We have revised the manuscript to shorten the sections on pathophysiology and echocardiographic markers, removing redundancies and improving clarity. In addition, we modified the structure to enhance the flow between sections, ensuring a more concise and cohesive narrative.
- The recommendation of the American Thoracic Society (ATS) and the American Heart Association (AHA) regarding the use of supplemental oxygen therapy for bronchopulmonary dysplasia associated pulmonary arterial hypertension is to control the target blood oxygen saturation (SaO2) between 92% and 95%. The recommended recommendation in this article is to maintain the SaO2 level of cPH patients at ≥92-95%, which is not a specific range. Clinically, there may be ambiguity regarding the low line value of SaO2 (i.e. choosing 92% or 95%).
Response: We thank the reviewer for this observation. To avoid ambiguity, we have revised the manuscript to align more closely with existing guidelines. Specifically, we now recommend maintaining SaOâ‚‚ in the range of 92–95%, consistent with ATS and AHA statements. The prior wording (“≥92–95%”) has been corrected to ensure clarity and precision for clinical application
- The conclusion is vague. It should precisely summarize the framework's key innovations and explicitly state the next steps for validation.
Response: We thank the reviewer for this helpful suggestion. We have revised the conclusion to provide a more precise summary of the framework’s key innovations, including the integration of updated evidence, standardized echocardiographic definitions, and consensus-based strategies for screening and management. We now also explicitly outline the next steps, highlighting ongoing implementation across Quebec NICUs and planned evaluation of feasibility, adherence, and impact to validate the framework.
While the intent of your work is valuable, the manuscript in its present form does not meet the bar for original research or a definitive consensus statement. I strongly encourage you to gather preliminary data on the application of your framework within the TnECHO-Qc network. Submitting a manuscript that demonstrates the real-world feasibility, acceptability, and initial outcomes of your algorithm would be a much stronger contribution to the literature.
Response: We appreciate the reviewer’s thoughtful feedback. We agree that demonstrating feasibility and outcomes will be an important next step and are actively implementing the framework across TnECHO-Qc sites with plans to collect and publish these data as follow-up work. At this stage, our goal was to provide a comprehensive, evidence-informed framework that integrates existing recommendations with updated evidence and provincial consensus, as a foundation for harmonized practice. We believe that publishing this framework now will support knowledge translation, foster adoption across NICUs, and facilitate the evaluation studies that are already in progress. Data regarding ongoing audits cannot be included in the manuscript at this stage because ethics approval was not obtained to publish these preliminary findings.
Reviewer 2 Report
Comments and Suggestions for Authors
Esteemed Authors,
I was honored to review the manuscript entitled" Targeted Neonatal Echocardiography in Bronchopulmonary Dysplasia: A Framework for Screening and Management of Chronic Pulmonary Hypertension".
The authors present the results of years of collaborative efforts by"neonatologists trained in targeted neonatal echocardiography (TnECHO), two neonatal nurse practitioners, and neonatal hemodynamic fellows actively engaged in training, working in four level-3 NICUs" in the Quebec area. In a well-structured paper, the authors describe the development in a long-term frame of the collaborative group. More importantly, the authors are sharing the criteria proposed for screening preterm infants at risk for chronic pulmonary hypertension following bronchopulmonary dysplasia. The presented algorithm delineates the proposed conduct for these infants based on risk, high-risk, and severity of the chronic pulmonary hypertension.
The most important part of the paper is dedicated to presenting the echocardiographic markers of chronic lung hypertension; special comments clarify all aspects of the selected sonographic indicators. The complex nature of the chronic lung condition is underlined, and he authors are proposing even a detailed phenotyping of the pulmonary circulation.
The authors are also grading the severity of the chronic lung hypertension and propose a follow-up rhythm based on the severity of the condition. Pro-BNP and NT-pro-BNP are cited as biomarkers, also underlying their limitations in the context.
The last part of the manuscript proposes a structured management protocol - non-pharmacological and pharmacological - for infants with chronic lung hypertension. I appreciate the details offered regarding each component of the protocol. My only concern is related to diuretics, as recent data, including a meta-analysis, failed to find a significant benefit of diuretics in BPD treatment. Please, see for example:
Ó Briain E, Byrne AO, Dowling J, Kiernan J, Lynch JCR, Alomairi L, Coyle L, Mulkerrin L, Mockler D, Fitzgerald G, Rehman LU, Semova G, Isweisi E, O'Sullivan A, O'Connor P, Mulligan K, Branagan A, Roche E, Meehan J, Molloy E. Diuretics use in the management of bronchopulmonary dysplasia in preterm infants: A systematic review. Acta Paediatr. 2024 Mar;113(3):394-402. doi: 10.1111/apa.17093. Epub 2024 Jan 12. PMID: 38214373.
In this context, I appreciate the citation of the paper by Baczynski M et al.
The conclusion aligns with the presented protocol and is clear.
Many congratulations to the authors on their collaborative work and well-written manuscript.
Author Response
October 30th, 2025
Dear Reviewers,
We would like to thank you for the careful evaluation of our manuscript entitled “Targeted Neonatal Echocardiography in Bronchopulmonary Dysplasia: A Framework for Screening and Management of Chronic Pulmonary Hypertension.”
We are grateful for the insightful comments and constructive suggestions provided, which have allowed us to clarify, refine, and strengthen our work. In this revised version, we have addressed all reviewer comments point by point and made changes to highlight the originality, methodological transparency, and clinical relevance of the proposed framework.
We appreciate the opportunity to improve our manuscript and look forward to your consideration.
Sincerely,
Dr Gabriel Altit, on behalf of the co-authors.
Dr Gabriel Altit, MDCM, MSc, FRCPC, FAAP
Neonatology – Montreal Children’s Hospital - McGill University
gabriel.altit@mcgill.ca
Reviewer 2:
Esteemed Authors,
I was honored to review the manuscript entitled" Targeted Neonatal Echocardiography in Bronchopulmonary Dysplasia: A Framework for Screening and Management of Chronic Pulmonary Hypertension". The authors present the results of years of collaborative efforts by"neonatologists trained in targeted neonatal echocardiography (TnECHO), two neonatal nurse practitioners, and neonatal hemodynamic fellows actively engaged in training, working in four level-3 NICUs" in the Quebec area. In a well-structured paper, the authors describe the development in a long-term frame of the collaborative group. More importantly, the authors are sharing the criteria proposed for screening preterm infants at risk for chronic pulmonary hypertension following bronchopulmonary dysplasia. The presented algorithm delineates the proposed conduct for these infants based on risk, high-risk, and severity of the chronic pulmonary hypertension. The most important part of the paper is dedicated to presenting the echocardiographic markers of chronic lung hypertension; special comments clarify all aspects of the selected sonographic indicators. The complex nature of the chronic lung condition is underlined, and he authors are proposing even a detailed phenotyping of the pulmonary circulation. The authors are also grading the severity of the chronic lung hypertension and propose a follow-up rhythm based on the severity of the condition. Pro-BNP and NT-pro-BNP are cited as biomarkers, also underlying their limitations in the context.
The last part of the manuscript proposes a structured management protocol - non-pharmacological and pharmacological - for infants with chronic lung hypertension. I appreciate the details offered regarding each component of the protocol. My only concern is related to diuretics, as recent data, including a meta-analysis, failed to find a significant benefit of diuretics in BPD treatment. Please, see for example: Ó Briain E, Byrne AO, Dowling J, Kiernan J, Lynch JCR, Alomairi L, Coyle L, Mulkerrin L, Mockler D, Fitzgerald G, Rehman LU, Semova G, Isweisi E, O'Sullivan A, O'Connor P, Mulligan K, Branagan A, Roche E, Meehan J, Molloy E. Diuretics use in the management of bronchopulmonary dysplasia in preterm infants: A systematic review. Acta Paediatr. 2024 Mar;113(3):394-402. doi: 10.1111/apa.17093. Epub 2024 Jan 12. PMID: 38214373. In this context, I appreciate the citation of the paper by Baczynski M et al. The conclusion aligns with the presented protocol and is clear. Many congratulations to the authors on their collaborative work and well-written manuscript.
Response: We thank the reviewer for the supportive comments and for highlighting this important point regarding the role of diuretics. We agree that the evidence supporting diuretic use in BPD, particularly in the context of chronic pulmonary hypertension, is limited and evolving. As suggested, we have added reference to the recent meta-analysis by Ó Briain et al. (2024) and clarified in the manuscript that the benefit of diuretics remains uncertain. In the revised text, diuretics are presented as a potential adjunctive therapy in select cases, with explicit acknowledgement of the lack of consistent evidence for long-term benefit.

Reviewer 3 Report
Comments and Suggestions for Authors
Comments to manuscript [Targeted Neonatal Echocardiography in Bronchopulmonary 2 Dysplasia: A Framework for Screening and Management of 3 Chronic Pulmonary Hypertension]
Dear authors, congratulations on the important initiative.
1- Elaborate on the originality of the review
2-Elaborate on the impact
3- As you mention: lines 64 to 66
[This manuscript describes this collaborative initiative that led to a province-wide standardized screening and management strategy for cPH associated with BPD in preterm infants, with a focus on the role of echocardiography in guiding clinical decision-making.] I suggest adding results from your experience with using this protocol
4-Regarding Figure 1
- The decision for [Consider for 28-30 weeks+ Respiratory support or O2at 36 weeks or BW <3rd percentile ] in the diagram of Figure 1 has no arrow to show what is the next step or to complete the line to join echo at 36 weeks
- It is not clear from the diagram if preterm infants develop PH before 36 weeks, and what steps to follow
5-Several paragraphs missed documentation and no reference to support, as Lines 114-135: no reference, lines 143 to 166, 171-222, and more. Authors need to document the reported data with suitable references
6- What are the limitations of the suggested protocol?
7- I hope to confirm the type of manuscript: is it a review article, /protocol /brief report
Author Response
October 30th, 2025
Dear Reviewers,
We would like to thank you for the careful evaluation of our manuscript entitled “Targeted Neonatal Echocardiography in Bronchopulmonary Dysplasia: A Framework for Screening and Management of Chronic Pulmonary Hypertension.”
We are grateful for the insightful comments and constructive suggestions provided, which have allowed us to clarify, refine, and strengthen our work. In this revised version, we have addressed all reviewer comments point by point and made changes to highlight the originality, methodological transparency, and clinical relevance of the proposed framework.
We appreciate the opportunity to improve our manuscript and look forward to your consideration.
Sincerely,
Dr Gabriel Altit, on behalf of the co-authors.
Dr Gabriel Altit, MDCM, MSc, FRCPC, FAAP
Neonatology – Montreal Children’s Hospital - McGill University
gabriel.altit@mcgill.ca
Reviewer 3:
Comments to manuscript [Targeted Neonatal Echocardiography in Bronchopulmonary 2 Dysplasia: A Framework for Screening and Management of 3 Chronic Pulmonary Hypertension]
Dear authors, congratulations on the important initiative.
- Elaborate on the originality of the review and 2-Elaborate on the impact
Response: We thank the reviewer for this helpful suggestion. We have revised the introduction and conclusion to better emphasize the originality and impact of our work. Specifically, we highlight that this is the first provincial consensus framework tailored to the NICU context, which merges existing international recommendations with updated evidence and harmonizes practice across multiple centers. We also underscore the anticipated impact of the framework: providing clinicians with a practical, standardized approach for screening and managing BPD-associated cPH, supporting knowledge translation, and laying the groundwork for future evaluation of feasibility, adherence, and outcomes.
3- As you mention: lines 64 to 66
[This manuscript describes this collaborative initiative that led to a province-wide standardized screening and management strategy for cPH associated with BPD in preterm infants, with a focus on the role of echocardiography in guiding clinical decision-making.] I suggest adding results from your experience with using this protocol
Response: We thank the reviewer for this thoughtful suggestion. At present, the framework has recently been implemented across participating centers, and data collection on feasibility, adherence, and clinical outcomes is underway. As such, outcome data are not yet available for inclusion in the current manuscript for all centers. We view this as an essential next step and plan to publish our experience with real-world implementation as a secondary follow-up work. In the current paper, our aim is to present the consensus-based framework itself as a foundation to harmonize practice and support subsequent evaluation.
Data from our ongoing audits could not be incorporated into the manuscript, as it originates from a Quality Improvement (QI) initiative for which Research Ethics Board (REB) approval was not obtained for publication of the numerical findings within a manuscript.
4-Regarding Figure 1
- The decision for [Consider for 28-30 weeks+ Respiratory support or O2at 36 weeks or BW <3rd percentile ] in the diagram of Figure 1 has no arrow to show what is the next step or to complete the line to join echo at 36 weeks
- It is not clear from the diagram if preterm infants develop PH before 36 weeks, and what steps to follow
Response: Thank you, we have modified Figure 1 accordingly
5-Several paragraphs missed documentation and no reference to support, as Lines 114-135: no reference, lines 143 to 166, 171-222, and more. Authors need to document the reported data with suitable references
Response: We thank the reviewer for pointing this out. In the revised manuscript, we have carefully reviewed all sections highlighted and added multiple appropriate references to support the statements.
6- What are the limitations of the suggested protocol?
Response: We thank the reviewer for this important point. We have added a section outlining the limitations of the proposed framework. These include reliance on expert consensus in areas where evidence remains limited, absence of validation data at this stage, and potential variability in applicability across different health care systems outside Quebec. We also acknowledge that some recommendations (e.g., use of diuretics, frequency of follow-up) are based on evolving evidence and may need refinement as new data emerge. Finally, feasibility and adherence will need to be confirmed through ongoing implementation and evaluation, which we plan to report in follow-up work.
7- I hope to confirm the type of manuscript: is it a review article, /protocol /brief report
Response: We thank the reviewer for this helpful point. We have clarified in the manuscript that this work represents a framework development manuscript. While it incorporates elements of a review by summarizing and integrating existing guidelines and recent evidence, its primary purpose is to present a consensus-based, province-wide framework for screening and management of BPD-associated cPH. We have revised the text to explicitly state the manuscript type to avoid ambiguity.

Round 2
Reviewer 3 Report
Comments and Suggestions for Authors
The authors responded adequately to all comments. I have no further comments.